# Visual Evidence for the Recruitment of Four Enzymes with RNase Activity to the *Bacillus subtilis* Replication Forks

**DOI:** 10.3390/cells13161381

**Published:** 2024-08-20

**Authors:** Rebecca Hinrichs, Peter L. Graumann

**Affiliations:** 1Centre for Synthetic Microbiology (SYNMIKRO), Philipps Universität Marburg, Karl-von-Frisch-Str. 14, 35043 Marburg, Germany; 2Department of Chemistry, Philipps-Universität Marburg, Hans-Meerwein-Straße 4, 35043 Marburg, Germany

**Keywords:** DNA replication, RNases, Okazaki fragment maturation, *Bacillus subtilis*, DNA polymerase 1, Exonuclease

## Abstract

Removal of RNA/DNA hybrids for the maturation of Okazaki fragments on the lagging strand, or due to misincorporation of ribonucleotides by DNA polymerases, is essential for all types of cells. In prokaryotic cells such as *Escherichia coli*, DNA polymerase 1 and RNase HI are supposed to remove RNA from Okazaki fragments, but many bacteria lack HI-type RNases, such as *Bacillus subtilis*. Previous work has demonstrated in vitro that four proteins are able to remove RNA from RNA/DNA hybrids, but their actual contribution to DNA replication is unclear. We have studied the dynamics of DNA polymerase A (similar to Pol 1), 5′->3′ exonuclease ExoR, and the two endoribonucleases RNase HII and HIII in *B. subtilis* using single-molecule tracking. We found that all four enzymes show a localization pattern similar to that of replicative DNA helicase. By scoring the distance of tracks to replication forks, we found that all four enzymes are enriched at DNA replication centers. After inducing UV damage, RNase HIII was even more strongly recruited to the replication forks, and PolA showed a more static behavior, indicative of longer binding events, whereas RNase HII and ExoR showed no response. Inhibition of replication by 6(p hydroxyphenylazo)-uracil (HPUra) demonstrated that both RNase HII and RNase HIII are directly involved in the replication. We found that the absence of ExoR increases the likelihood of RNase HIII at the forks, indicating that substrate availability rather than direct protein interactions may be a major driver for the recruitment of RNases to the lagging strands. Thus, *B. subtilis* replication forks appear to be an intermediate between *E. coli* type and eukaryotic replication forks and employ a multitude of RNases, rather than any dedicated enzyme for RNA/DNA hybrid removal.

## 1. Introduction

RNA maturation, RNA degradation, and RNA turnover are essential processes for all kinds of cells. Enzymes known as ribonucleases (RNases) are crucial for these processes and are usually present in many different versions per cell [1,2,3,4]. During DNA replication, RNases play a major role in the maturation of Okazaki fragments on the discontinuous, lagging strand, by removing the arising DNA/RNA hybrids made by DNA primase [5,6]. In addition, even though DNA polymerases have a much higher affinity for deoxyribonucleotides (dNTPs) than ribonucleotides, the ribonucleoside triphosphate (RTP) pools are in large excess of those of dNTPs. They can be incorporated into the growing primer strand and are thus present in the newly synthesized strand due to misincorporation. Therefore, in vitro data show that ribonucleo-monophosphates (rNMPs) are incorporated about every 2.3 kb in *E. coli* cells. Loss of removal of rNMPs leads to an increase in mutation frequency [7].

A general differentiation can be made between endoribonucleases and exoribonucleases. RNase E or RNase Y are major endoribonucleases that initiate RNA turnover, which is then taken over by exonucleases [8]. The latter process RNA molecules that result in 3′ or 5′ terminal release of nucleotide residues. A 3′-5′ exoribonuclease in *Bacillus subtilis* is exemplified by polynucleotide phosphorylase (PNPase), while magnesium-dependent ExoR (also called FenA) and RNase J1 are 5′-3′ exonucleases [8,9]. In addition to the classical ribonucleases, several other enzymes also exhibit exonucleolytic functions on RNA/DNA hybrids, like the DNA polymerase Pol I or PolA, which exhibit 5′-3′ exonuclease activity in *E. coli* or *B. subtilis*, respectively [10]. Pol I and PolA are thought to contribute to RNA removal at replication forks in bacteria, in conjunction with or in addition to different RNases [11]. During DNA replication in eukaryotes, one RNase was discovered, which belongs to the RNase H family: RNase H1 hydrolyzes RNA from RNA/DNA hybrids [12]. Prokaryotes possess one or two of three H-type RNases, RNase HI, HII, or RNase HIII. In *E. coli*, RNase HI hydrolyzes RNA–DNA hybrids, which contain polymers of four or more ribonucleotides [13]. RNase HII is characterized by hydrolyzing at the 5′ to a single ribonucleotide; i.e., it acts as an endonuclease, unlike RNase HI. Also, it hydrolyzes at the 5′ to the ribonucleoside monophosphate [7]. Thus, it is important to initiate the removal of incorporated rNMPs from the genome, to ensure genomic integrity [14]. RNase HIII has a high enzymatic similarity to RNase HI [15]. Bacteria are considered to generally have two RNase H enzymes, which in *Bacillus subtilis* are RNase HII and RNase HIII [16]. Most endoribonucleases cleave RNA in the presence of divalent cations, resulting in fragments containing 3′-hydroxyl and 5′-phosphate termini [9], including RNase HII and RNase HIII.

During DNA replication in *B. subtilis*, continuous synthesis of the leading strand occurs by polymerase PolC. For lagging strand synthesis, primase DnaG synthesizes RNA primers, which are extended by DNA polymerase DnaE [17,18]. DnaE is thought to only extend by a few bases and then hand over to PolC [19], which finishes the 1–2 kb Okazaki fragments [20]. Indeed, 2000–4000 fragments per 4.6 Mb chromosome are mentioned in *E. coli* for whole chromosome replication [21]. Removal of RNA primers is important for DNA stability [22] because the lagging strand synthesis is terminated by ligase LigA sealing remaining single-strand gaps [23]. As pointed out above, DNA polymerase PolA shows 5′-3′ exonuclease activity and could be detected together with RNase HIII and ExoR in in vitro studies for the maturation of Okazaki fragments in *B. subtilis* [24]. Also, as stated above, RNase HII appears to be responsible for the removal of individual rNMPs that are incorporated into DNA by DNA polymerase during DNA replication [7,25]. Similarly, PolA, as well as ExoR, could be involved in the repair of DNA damage due to UV [26]. Interestingly, temperature-sensitive phenotypes have been discovered in different deletions combinations. It has been shown that the deletion of *rnhC* and *polA*, as well as the deletion of *rnhC* and *exoR*, lead to lethality under 25 °C growth conditions [24], while a double mutation (*rnhB*, *rnhC*) leads to poor growth [7]. These findings point to a possible role of these proteins at *B. subtilis* replication forks, but other explanations for synthetic phenotypes are possible. In order to obtain definite proof for enzymatic activity of the four mentioned proteins at replication forks, or at sites all over the chromosome distinct from replication events, we monitored single-molecule dynamics, to test if replication forks represent sites of frequent stops for different enzymes having RNase activity.

## 2. Materials and Methods

### 2.1. Strain Construction

For the construction of the mV/mNeo fusions in *B. subtilis*, the integration plasmid pSG1164 was used [27]. Using pSG1164, the corresponding fluorophore is integrated through a single crossover c-terminal to the original locus [28]. For integration, a 500 bp homolog of the C-terminus of the gene of interest must be cloned using Gibson assembly [29] into the vector next to the linker and mV/mNeo sequence. Used primers have a homolog 25 bp overhang and are listed in Appendix A. The transformed plasmids were extracted using a kit (New England BioLabs, Frankfurt am Main, Germany). The deletion strains are based on *B. subtilis* BG214 and contain RNase HIII-mV or ExoR-mV constructs and were generated by transformation of cells with chromosomal DNA from *B. subtilis* 168 Δ*rnhC::kan trpC2*, Δ*exoR::kan trpC2*, obtained from the *Bacillus* Genetic Stock Center (Columbus, OH, USA) [30]. The chromosomal DNA was extracted using a kit (innuPREP Bacteria DNA Kit, Analytik-Jena, Jena, Germany).

### 2.2. Growth Conditions

All plasmids, strains and oligonucleotides used are listed in Appendix A. *E. coli* strains were cultured in LB (lysogenic broth) medium at 37 °C. Likewise, LB medium was used for microscopy (SMT). For growth curves (Appendix A), S750 minimal medium was used (100 mL: 10 mL 10 × S750 salt solution [1 L ddH_2_O; pH 7.0: 104.7 g MOPS, 13.2 g of (NH_4_)_2_SO_4_, 6.8 g of KH_2_PO_4_, 12 g of KOH], 1 mL 100 × metal solution [100 mL: 20 mL of MgCl_2_ (1 M), 7 mL of CaCl_2_ (1 M), 0.5 mL of MnCl_2_ (1 M), 1 mL of ZnCl_2_ (0.1 M), 1 mL of FeCl_3_ (50 mM), 5 mL of Thiamine hydrochloride (2 mg/mL), 17 µL of HCl (2 M)], 2 mL 50% fructose (*w/v*), 1 mL of 10% L-glutamate (*w/v*), 40 µL of 1% casamino acids (*w/v*) [31]. For this experiment, the *B. subtilis* strains were cultured in LB at 37 °C. When needed, antibiotics were added at the following concentrations: ampicillin 100 µg/mL, chloramphenicol 5 µg/mL, kanamycin 30 µg/mL, 6(p-hydroxyphenylazo)-uracil (HPUra) 15 mg/mL; 0.5% xylose was added from a 50% sterile filtrated stock solution in ddH_2_O.

### 2.3. Western Blot

The samples were harvested from the exponential growth phase and digested by lysozyme. The detection was performed with a primary polyclonal α GFP-tag antibody (1:5000) (SAB4301795, Sigma-Aldrich, Merck KGaA, Darmstadt, Germany)/mNeonGreen-tag (1:4000) (E8E3V, Cell Signaling Technology, Beverly, MA, USA) and secondary antibody goat-anti-Rabbit-IgG, peroxidase-conjugated (1:100,000) (Sigma-Aldrich). A colour prestained protein standard (New England BioLabs, Frankfurt am Main, Germany) was used.

### 2.4. Fluorescence Microscopy

Microscopy was performed in LB medium with a prior cultivation of 30 °C, 200 rpm. Cells were analysed in the exponential growth phase. For wide-field epifluorescence microscopy, a Zeiss Observer A1 microscope (Carl Zeiss, Oberkochen, Germany) with an oil immersion objective (100 × magnification, 1.45 numerical aperture, alpha PlanFLUAR; Carl Zeiss) was used. The images were recorded with a charge-coupled-device (CCD) camera (CoolSNAP EZ; Photometrics, AZ, USA) and an HXP 120 metal halide fluorescence illumination with intensity control [32]. For the sample preparation, a round coverslip (25 mm, Marienfeld, Lauda-Königshofen, Germany) was used and covered 5 µL of cell culture with a 1.5% agarose pad. The agarose pads were made with water by sandwiching 100 µL of the melted agarose between two smaller coverslips (12 mm, Menzel, Braunschweig, Germany). Images were processed using ImageJ (1.5.4a) [32].

### 2.5. Single-Molecule Tracking (SMT)

Individual molecules were tracked using a custom-made slim-field setup on an inverted fluorescence microscope (Nikon Eclipse Ti-E, Nikon Instruments Inc., Tokyo, Japan). An EMCCD camera (ImagEM X2 EM-CCD, Hamamatsu Photonics KK, Hamamatsu, Japan) was used to ensure high-resolution detection of the emission signal, resulting in a calculated resolution of the position of the molecule down to 20 nm. The central part of a 514 nm laser diode (max power 100 mW, TOPTICA Beam Smart, Toptica, Munich, Germany) was used with up to 20% of the intensity (about 160 W cm^−2^ in the image plane) to excite samples, fused to mNeonGreen (using laser filter set BrightLine 500/24, dichroic mirror 520 and BrightLine 542/27) by focusing the beam onto the back focal plane of the objective. A CFI Apochromat objective (TIRF 100 × Oil, NA 1.49) was used in the setup [33]. For the analysis, a video of 3000 frames at 20 ms was recorded, of which 1000, starting after 200 to 300 frames, dependent on the time point when single-molecule levels were reached due to bleaching, were used for the analysis. Software Oufti 1.0 [34] was used to set the necessary cell meshes. Utrack [35] was employed for the automatic determination of molecule trajectories. Data analysis was carried out using software SMTracker 2.0 [33,36].

## 3. Results

### 3.1. PolA, ExoR, and RNase HII Show Nucleoid Localization

To investigate the localization of PolA, ExoR, RNase HII, and RNase HIII in vivo, C-terminal mVenus or mNeonGreen (mNeo) fusions were created and integrated at the original locus. The localization of the replication forks was visualized using a DnaX-CFP (DNA polymerase III, part of the clamp-loader complex) allele in the same strain. To ensure that the fused proteins are expressed at full length, a Western blot was performed against the corresponding fluorophore (Appendix A): the experiments showed expression of full-length protein fusions in all cases. Of note, restreaking of the RNase HIII-mNeo strain resulted in the loss of the chromosome-integrated plasmid, for a reason unknown to us, because the RNase HIII (*rnhC*) deletion strain does not show a detectable phenotype at 37 °C growth conditions. We therefore performed all experiments using freshly transformed cells showing full-length expression of fusion proteins (Appendix A). The functionality of all strains was tested in a growth curve under different conditions (Appendix A). *RnhC*/*rnhB* double mutant cells show a strong growth defect, but RNase HII-mNeo Δ*rnhC* cells or RNase III-mNeo Δ*rnhB* cells grew like wild type cells (Appendix A). Likewise, *polA exoR* double mutant cells grew very poorly, while the PolA-mNeo or ExoR-mNeo fusions combined with the respective deletion did not show any defect (Appendix A).

Using epifluorescence microscopy (Figure 1), we found that ExoR-mNeo, PolA-mNeo, and RNase HII-mNeo show a clear staining pattern comparable to a DAPI stain (Figure 1A,B), and thus a localization pattern to the nucleoids in the cell [37]. Contrarily, the fusion of RNase HIII showed a diffuse localization throughout the cells (Figure 1B). The CFP channels provide information about the localization of the replication forks: the DnaX-CFP fusion showed distinct foci in the cells (1–2 per cell). The merge of both channels shows the colocalization of the proteins relative to DnaX, revealing a clear overlap of signals, as expected (Figure 1B, yellow). Localization of PolA, ExoR, and RNase HII to nucleoids is likely due to non-specific binding to chromosomal DNA and thus diffusion through the nucleoids, masking any colocalization with replication forks using epifluorescence.

### 3.2. Single-Molecule Tracking Reveals Distinctive Patterns of Motion for RNases

Epifluorescence experiments did not reveal an association of any RNase or PolA with replication forks. To obtain a better understanding of protein dynamics and increased spatiotemporal resolution, we used single-molecule tracking (SMT) [33]. The method employs a beam of a 514-nm laser diode, which is expanded by a factor of 20, and the central part is focused on the rear focal plane of the 100 × A = 1.49 objective. SMT allows for visualization of events of molecules located at a defined subcellular location with an accuracy of 40 nm or less [36]. SMT was performed with 20 ms stream acquisition. To determine the area of the cell to be detected, cell meshes were set by using Oufti [34] and trajectories were determined by u-track [35]. Tracks of only 5 steps and more were used. Analysis of the resulting data, all from biological triplicates, was carried out with the SMTracker 2.0 [33].

Figure 2A shows the heat maps of the given proteins to visualize the localization of molecule tracks in the cell. For this, we projected all tracks from the biological replicates into a cell with an average size of 3 × 1 µm. The distribution of tracks is indicated by a colour shift from yellow (low probability) to black (highest probability). The intensity of the maps created is adapted to each other. To be able to compare localization patterns with already known replication proteins, a fusion of DnaC (DNA helicase) was included [38]. To investigate a protein without known involvement in replication, YaaT (part of the Y-complex) was included. YaaT, is a component of the Y-complex and possible specificity factor for RNase Y within the degradosome [39]. It shows strong membrane association, but is also found statically bound within the cytoplasm [40]. Similar to the epifluorescence images (Figure 1), the preferred sites of localization of PolA, ExoR and RNase HII tracks is clearly seen on the nucleotides, with a concentration to the central parts of the nucleoids, very similar to DnaC. In contrast to epifluorescence imaging, the preferred localization of RNase HIII tracks is clearly on the nucleoids and is most similar to that of DnaC (Figure 2A). These data support the notion that RNase HII and HIII are associated with removal of RNA/DNA hybrids from the DNA and may be associated with replication.

We next determined the diffusion constants of the proteins using squared displacement analyses (SQD), shown in Figure 2B,C. SQD analysis can be used to determine the average diffusion constant of a molecule, as well as to analyse if a single or several populations with different diffusion constants exist; if several, the size of the populations can be determined. For PolA, ExoR and RNases HII and HIII, a two-population Rayleigh fit was used, which explained the observed distribution of tracks very well. This analysis suggests the existence of a slow mobile/static and a high mobile population, most likely freely diffusive molecules. The bubble blot (Figure 2C) visualizes diffusion constants and sizes of the two assumed populations, revealing that all proteins have slow-mobile fractions whose diffusion constants are in a similar range, one that has been described for tight DNA-binding events. In the case of DnaC, this is hexamer formation ahead of replication forks. Based on its high processivity at the forks, DnaC shows the lowest diffusion constant; higher constants of the slow-mobile fractions of PolA and RNases may indicate more transient binding to substrate sites. ExoR, RNase HII and DnaC have the largest slow mobile/static fractions, while RNase HIII has the largest diffusive population (Figure 2C). Free diffusion of RNase HIII monomers may explain why epifluorescence microscopy shows diffuse RNase HIII-mNeo localization (Figure 1B), hiding the transient binding at chromosomal sites that only become visible using SMT (Figure 2A). The control strain YaaT-mNeo showed a large static fraction (Figure 2C), which is partially based on the binding to the membrane (RNA degradosome), but also on binding to sites on the nucleoids [40]. Thus, a quarter to a third of molecules appear to be engaged in a substrate-bound form. For detailed numbers, please also consult Table 1. Although informative, these data still do not show if RNase activity of the studied proteins is associated with replication forks in vivo, other than for PolA and ExoR, where spatial connection to replication has been shown before [26].

### 3.3. All Four Enzymes Showing RNase Activity Feature Close Spatial Proximity of Motion and Frequent Arrests at Replication Forks

There is a convincing argument that an enzyme takes part in a reaction at a defined subcellular space in proximity of motion at or close to that site. We used a tool in SMTracker 2.0 that allowed us to score proximity of molecule trajectories close to sites in the cell that can be defined, e.g., by localizing a protein complex using a protein fusion having a different fluorescent colour. Figure 3A shows an example of two cells in which the position of replication forks has been determined by acquiring an image in the CFP channel (indicated by the stars), detecting DnaX-CFP, a component of the clamp loading complex. Blue tracks reveal freely diffusive trajectories of RNase HII-mNeo, and red tracks show trajectories staying within a radius of 106 nm for at least 5 time points, i.e., molecules showing confined diffusion, likely due to binding events. Several of these are close to DnaX-CFP signals and thus close to or at replication forks. Green trajectories reveal events of transition between free diffusion and confined motion of RNase HII-mNeo, logically being close to events of confined motion. Thus, our analyses can capture events of transitions from free diffusion to DNA binding and release from DNA binding. The lower right cell in Figure 3A likely contains a second replication that was not captured in the CFP channel fork, close to two events of confined motion for RNase HII. Such events were not counted as “close to replication forks”, while confined tracks overlapping with DnaX-CFP foci were considered as binding events to forks.

To obtain a general overview of proximity of tracks to replication forks, SMTracker was used to determine the probability of tracks being close to DnaX foci. Figure 3B shows a peak of DnaC-mNeo tracks relative to DnaX-CFP at about 250 nm, corresponding to the resolution of epifluorescence light microscopy (DnaX-CFP foci). The distribution of DnaC-mNeo tracks is roughly corresponding to a Rayleigh distribution. The peak at 250 nm (rather than at “0”) is also due to a change in filters and illumination between taking the CFP image by epifluorescence and by tracking DnaC-mNeo molecules by SMT, because replication forks are quite mobile [41]. In contrast to DnaC-mNeo, the distribution of YaaT-mNeo showed two peaks, one close to 500 nm (events of YaaT binding on the nucleoids) and one at 1.8 µM, likely binding events at the cell membrane.

PolA-mNeo, ExoR-mNeo, RNase HII-mNeo and HIII-mNeo showed a single Rayleigh distribution with a peak of around 400 nm (Figure 3C–F). Because this general proximity is much less pronounced than for DnaC, we treated cells with UV, for which a stronger recruitment of PolA and ExoR to replication forks has been shown [26]. UV treatment resulted in a pronounced shift of RNase HIII and PolA to positions closer to replication forks; for ExoR and RNase HII, the shift was very mild but noticeable. These data show that PolA, ExoR and RNase HII and HIII show high probability of motion close to sites of DNA replication, especially under conditions of frequent replication fork arrest. These analyses do not quantify the extent of a change in motion, i.e., frequency of arrest, at replication forks. We searched for further evidence for the presence of four enzymes with RNase activity by confinement analysis in relation to the replication forks (Figure 4). For this, we scored for confined tracks, i.e., tracks that stay within three times the localization precision of this work, for 5 or more time points. We analysed only cells with distinct DnaX-CFP signals and counted how many cells had confined tracks immediately at a replication fork.

We used a tracking time of 1000 frames; therefore, confinement analyses show the percentage of cells having molecules arresting at DnaX-CFP foci within a 20 s time frame. Interestingly, all enzymes (PolA (71.4%), ExoR (72.7%), RNase HII (76%) and RNase HIII (72.1%) showed confined tracks at replication forks in more than 70% of cells (Figure 4C). As a reference, replicative helicase showed a similar percentage of more than 78% confinement at forks, whereas for the negative control, YaaT-mNeo, only around 8% were counted. These data strongly suggest that Okazaki fragment maturation involves the recruitment of at least four enzymes to the replication forks in *B. subtilis*. Because we wished to obtain additional evidence for this idea, we tested if enzymes would be more strongly engaged as replication forks became stalled, e.g., in the response to UV irradiation, which of course also induced DNA repair events at many other sites on the chromosome.

### 3.4. PolA, but Not RNase HIII or HII, Shows a Change in Mobility in Response to UV Light-Induced DNA Damage

To test the influence of DNA damage on the dynamics of RNases HII and HIII and to identify a possible involvement in the UV repair system, the dynamics of the proteins under UV-damage of DNA were determined. Therefore we induced DNA damage via crosslinks using UV light [10], using a treatment of 60 J m^−2^. Please note that for this analysis, diffusion constants for each protein were determined by a simultaneous fit, in order to better compare changes in population size. Interestingly, only the population sizes of PolA-mNeo changed significantly. After UV treatment, there was an increase in the static population from 30.2% to 43.2%, with a concomitant decrease of the high-mobile fraction (Figure 5, Appendix A). ExoR-mNeo, RNase HII-mNeo and RNase HIII-mNeo did not show any significant changes in their dynamics in response to UV treatment. This is somewhat contradictory to earlier reports of our group, in which significant changes were also detected for ExoR [26]. We will come to this point later. The lack of changes in RNase HII and HIII dynamics suggest that the RNases are not directly involved in the repair of UV damage, like PolA. However, because we observed a pronounced shift of RNase HIII towards replication forks after UV damage induction (Figure 3D) and a mild effect on HII (Figure 3F), our findings suggest that replication forks that are dealing with UV damage produce more errors in the incorporation of ribonucleotides, as a substrate for RNases.

### 3.5. Lack of ExoR Affects RNase HIII Dynamics

Previous work has shown that double mutation of *rnhC* and *exoR* results in a phenotype that is lethal at 25 °C [24]. We investigated a possible influence of the proteins on each other’s activity by analysing the dynamics of each in strains carrying the respective deletion of the other gene (ExoR-mVenus Δ*rnhC*, RNase HIII-mV Δ*exoR*). In cases where both proteins are recruited to replication forks via specific protein/protein interaction with proteins present at the forks at all times, we would not expect strong alterations in, e.g., static populations representing DNA-bound molecules, while independent recruitment due to substrate availability might lead to detectable changes in DNA-bound, slow mobile/static states.

For these analyses, we projected all tracks showing confined motion from the three biological replicates into an average size cell of 3 × 1 μm size (“confinement heat map”, *B. subtilis* cells are on average 0.75 μm wide and 2 to 4 μm long). To achieve this, tracks were sorted into those that stayed within a radius of 120 nm, determined as three times our localization error, for at least 6 consecutive steps (confined motion), and into those that showed large displacements, indicative of free diffusion [40]. For ExoR-mV, as well as RNase HIII-mV with and without deletions, there were no significant differences in the pattern of confined tracks (Figure 6A). For ExoR-mV, the static fraction decreased from 47.7% 410 to 36% in the deletion background of *rnhC*, compared to wild type cells (Figure 6B). Conversely, RNase HIII-mV Δ*exoR* cells showed an increase in the static population (43.3% to 51.3%) (Figure 6B and Table 1).

### 3.6. Inhibiting PolC Activity Leads to a Strong Effect on the Localization of RNase HII and HIII

Because there were no detectable changes in the single-molecule dynamics of RNase HII-mNeo or RNase HIII-mNeo during UV stress (Figure 5), we employed a third strategy for testing association with replication forks. We employed treatment to arrest DNA replication via inhibition of DNA polymerase PolC, using 6(p-hydroxyphenylazo)-uracil (HPUra), which reversibly binds to and inhibits DNA polymerase PolC. Binding completely blocks the progression of replication forks [42], as opposed to slowing down replication due to the necessity of repairing base-dimers in response to UV irradiation. Figure 7A shows corresponding heat maps for RNase HII-mNeo and RNase HIII-mNeo and DnaC-mNeo. As shown in Figure 3, both “H” RNases localize mainly to the nucleoids and much less so to sites surrounding the nucleoids. Note that the heat maps are somewhat different because normal growth conditions and HPUra stress conditions were equally scaled. Following PolC inhibition, RNase HII did not change its localization pattern, while RNase HIII became more dispersed, still maintaining some strong localization to a subpolar site. In contrast to these marked changes, DnaC did not change much in its localization pattern (Figure 7A). Based on the idea that *B. subtilis* replication forks do not disintegrate when blocked [43], DnaC also maintained an about 50/50 ratio of freely diffusing/statically bound molecules after addition of HPUra (Figure 7B, Table 1). This was different from RNases: RNase HII became less statically engaged (static fraction from 43 to 30% after HPUra), while RNase HIII showed a considerable increase in its static fraction (36 to 46%). Possibly, RNase HII has fewer substrate sites on its DNA due to a lack of incorporation of NTPs by PolC. For RNase HIII, an explanation for the increase in binding is less straightforward unless RNase HIII would be actively recruited to blocked replication forks, for which there is no evidence as yet.

In any event, a block in replication fork progression alters the single-molecule dynamics of both type H RNases, strongly supporting the idea that they are intimately involved in the processing of Okazaki fragments and removal of misincorporated NTPs.

## 4. Discussion

Incorporation of short stretches of RNA as primers for lagging strand synthesis requires the activity of enzymes that remove RNA from DNA/RNA hybrids in all organisms. In eukaryotic cells, RNase H2 and Exo1, together with the flap endonucleases Fen1 and Dna2, have been reported to be involved in Okazaki fragment maturation [44], while in *E. coli*, it is believed that DNA Pol I and RNase HI take over this task [11]. In the Gram-positive model organism *B. subtilis*, RNase HIII, exonuclease ExoR and DNA polymerase PolA have been proposed to be involved in replication, based on biochemical and genetic data [24]. It has also been suggested that RNase HIII acts as an important supporter of PolA in the maturation of Okazaki fragments [24]: it has been shown that RNase HIII influences endonucleolytic cleavage of RNA in RNA–DNA hybrid molecules in the processing of R-loops and maturation of Okazaki fragments. Additionally, the absence of both H-type RNases, HII and HIII, leads to a synthetic slow-growth phenotype [7], indicating that RNase HII may also be involved. However, the exact in vivo function of RNase HII is not known except for the possible removal of single rNTPs incorporated into DNA by DNA polymerases (DNA replication [7,45]. We sought to provide in vivo evidence to answer the question of which enzymes are truly involved in RNA removal from replication forks. Using single-molecule tracking, we could show that RNase HII and RNase HIII, as well as ExoR and PolA, are arrested at replication forks with a frequency that is close to that of molecules of replicative helicase DnaC. In response to DNA damage induced via UV stress, we found increased activity of RNase HIII and Pol A at the forks and different molecule dynamics during replication fork arrest or in mutant strains, all supporting a direct involvement of four enzymes in rNMP removal at the forks.

Single-molecule trajectories for all four enzymes could be explained by assuming two populations having distinct diffusion constants. Based on the idea that DNA polymerases or RNases detecting DNA/RNA hybrids would be in a bound state, where there is little diffusion (basically that displayed by DNA strands), the slow mobile population should represent enzymatically active enzymes bound to DNA, while the highly mobile fraction should represent diffusing molecules. Diffusion constants for all proteins were quite low in comparison to freely diffusing enzymes [46], and PolA, ExoR and RNase HII showed clear nucleoid staining in epifluorescence microscopy. Heat maps of all single-molecule trajectories showed a clear enrichment of tracks at central places, even within nucleoid areas, similar to those of DNA polymerase C, and very different from the more peripheral pattern of YaaT, which is known as a specific factor for the RNase Y in RNA degradation within the membrane-associated RNA degradosome. These data show that like DNA transcription factors [47], or sequence-specific DNA methyltransferases [48], H-type RNases and ExoR mostly employ constrained motion through the nucleoids, diffusing between DNA strands to find binding targets, which would explain the rather low mobility of the fast-mobile fraction.

RNase HII and HIII belong to the RNase H enzyme family, which is responsible for the identification and cleavage of RNA–DNA hybrids [12,13]. The formation of R-loops and the resulting DNA–RNA hybrids is a known stress response. This stress can be caused by cell wall damage, osmotic stress and oxidative damage, but also DNA damage, and can lead to genomic instability and replication arrest [49,50]. Using confinement analysis relative to the replication forks, we were able to clearly show that all four analysed enzymes have a very similar percentage of confined tracks in the direct proximity of the replication fork, similar to DNA helicase (DnaC). Confined tracks show dwell events for at least 100 ms, likely reflecting enzymatic activity in DNA-bound form. Based on distance determination between molecule trajectories and replication forks, we could clearly see an abundance of RNase HIII at the replication forks. Interestingly, under the influence of DNA damage (UV), RNase HIII became more enriched at sites close to the forks, possibly reflecting its published involvement in R-loop processing in *Bacillus subtilis* [51]. RNase HII also showed a high degree of spatial proximity to sites of DNA replication, in a range similar to that of RNase HIII, suggesting it may be also involved in Okazaki fragment maturation: Okazaki fragment length is 1000 to 2000 bp in bacteria, and erroneous incorporation of RNA nucleotides by DNA polymerases is in a similar range. Although generally, DNA polymerases prefer NTPs over rNTPs about 10,000-fold due to a “steric gate”, whereby a bulky side chain clashes with the 3′OH group of incoming ribonucleotides [52,53,54] and NTPs are in 10- to 100-fold excess over dNTPs [55,56]; thus, NTPs become incorporated about every 2300 Kb in yeast cells and similarly in bacterial cells [7]. In contrast to RNase HIII, HII did not show any stronger shift to the forks after DNA damage was induced by UV. PolA showed a stronger engagement with forks following UV irradiation, as was reported before [26]. In contrast to this study, we found PolA even more strongly associated with DNA replication sites even before the induction of DNA damage and similarly for ExoR. We believe that these differences are due to larger sample sizes in this work, showing significant localization even during normal growth. Likewise, the mNeo fluorophore used can lead to fewer artifacts, as it has a longer lifetime, higher brightness and shorter maturation time in contrast to mVenus [57]. The improvement of the fluorophore in combination with the higher sample size can lead to deviations.

In order to use a third means to show involvement of RNase HII and HIII, we tracked the fusion proteins during a replication block by HPUra, where the activity of PolC is inhibited. We observed strong differences in mobilities and localization pattern. RNase HII became more dynamic whereas RNase HIII showed more static (substrate-bound) motion. For RNase HIII, we observed a strongly altered localization is response to a block in replication. Overall, our in vivo analysis provides evidence that both proteins are actively involved at replication forks. Similarly, only RNase HIII, together with DNA polymerase I (PolA), seems to show significant responses to UV stress, underlining the already assumed interdependence of the two proteins. It could be shown that RNase HIII supports PolA in the maturation of Okazaki fragments [24]. Interestingly, RNase HII, as well as ExoR, shows very similar dynamics and localizations in the cell. RNase HII is known to remove single ribonucleoside monophosphates (rNMPs) during DNA replication [7]. This function is important for the cell because DNA is more stable than RNA. Resulting rNMP residues in DNA could lead to spontaneous strand breaks. ExoR and endoribunclease RNase HII showed very similar dynamics and localization, but it is unclear if they have similar functions, in line with their synthetic lethality with PolA or RNase HIII, respectively [58]. However, RNase HII and HIII treated with HPUra demonstrated a response to replication blockage. Thus, the single-molecule tracking of these treated strains has revealed that although RNase HII and RNase HIII appear to have distinct functions in vivo, the analysis suggests that RNase HII may have a supporting effect on the maturation of Okazaki fragments and might be a RNase involved in this process.

We also addressed the question of whether enzymes may be specifically recruited to forks in order to remove RNA primers, or whether these sites are found by a diffusion/capture mode. We therefore analysed the dynamics of ExoR and RNase HIII in the corresponding deletion backgrounds [24]. The dynamics of RNase HIII revealed a remarkable increase in the static population in the absence of *exoR*, while ExoR showed a slight decrease in the static population in the deletion background of *rnhC*. Thus, the lack of ExoR leads to mode binding events of RNase HIII, based on its higher abundance in the low-mobility state, apparently leading to a takeover of RNA nucleotide removal by RNase HIII. These data suggest that RNAse HIII shows a diffusion/capture mode of recruitment at replication forks, besides possible, unknown direct protein/protein interactions.

With the help of previous work [7,24,27,45,59] and our in vivo analyses, we conclude that *B. subtilis* has replication forks that strongly deviate from textbook knowledge on bacterial replication forks. Initiation of replication leads to loading and ensuing unwinding activity of the double-stranded DNA by the DNA helicase DnaC. The leading strand (5 to 3 direction) is then continuously processed by DNA polymerase C holoenzyme. For the replication of the lagging strand (3 to 5 direction), discontinuous processing occurs (Figure 8). Small RNA primers start each Okazaki fragment, which is synthesized by the primase (DnaG). DNA polymerase E, extends RNA primers by some DNA bases, and hands over synthesis of the lagging strand to DNA PolC (Figure 8). DNA ligase (LigA) ligates all fragments of the lagging strand at the 3′ end. At least four specific exo- or endonucleases are used to remove RNA primers from the lagging strand. A known one is DNA polymerase I (PolA), which removes these primers with its exonuclease function (5′-3′ exonuclease activity) together with the endoribonuclease RNase HIII [24]. In a likely redundant manner, RNaseH II and ExoR also remove RNA primers, while only RNase HIII and PolA show increased recruitment are suitable for UV damage and could be a part of the SOS response in *B. subtilis* [60]. However, it can be assumed that RNase HIII has functions outside of replication forks, because it shows areas of constrained motion at many sites on the nucleoids (Figure 6A).

Altogether, our work clarifies functions for four enzymes removing DNA/RNA hybrids from replication forks, which may be a unique multitude of proteins for *B. subtilis*. It will be interesting to investigate if some or many other bacterial species also distribute RNA removal during replication onto many enzyme’s shoulders.

## Figures and Tables

**Figure 1 cells-13-01381-f001:**
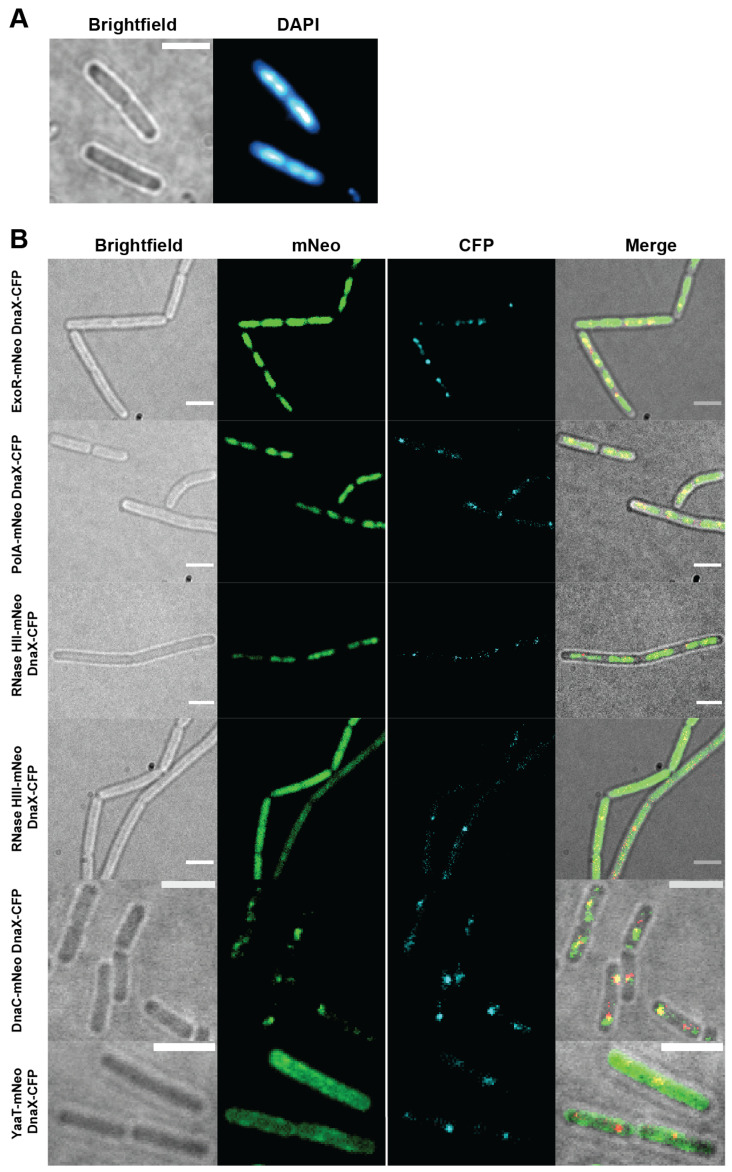
Epifluorescence microscopy of fusion strains. Panel (**A**) shows an example of an DAPI stain to visualize the chromosome in the cell. Panel (**B**) shows the brightfield images, the mNeonGreen (mNeo) channel of the fusions, DnaX-CFP for replication fork localization and a corresponding merge. Scale bar 2 µm.

**Figure 2 cells-13-01381-f002:**
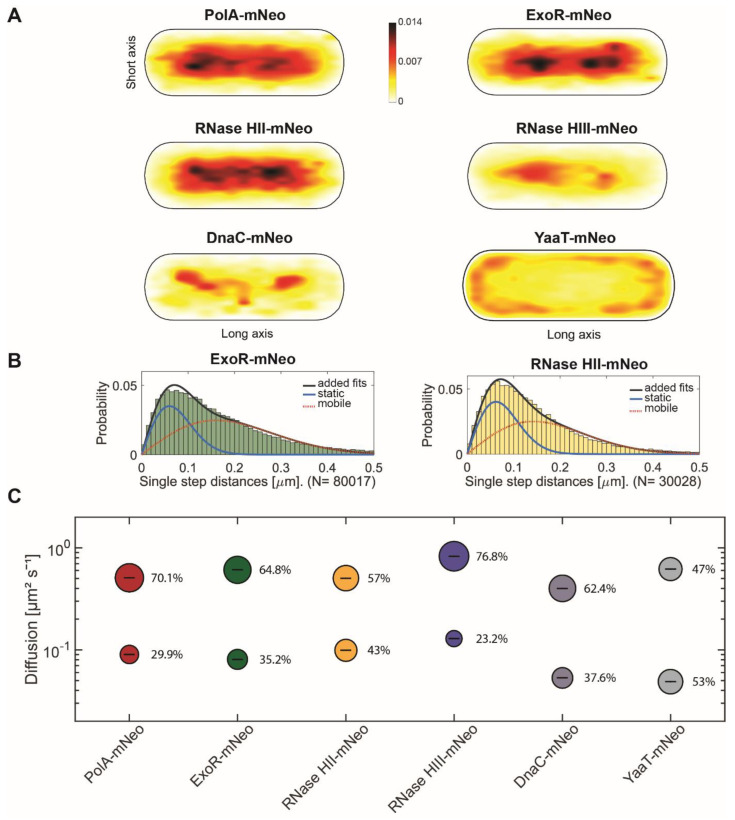
Single-molecule analyses of PolA-mNeo, ExoR-mNeo, RNase HII-mNeo, RNase HIII-mNeo, DnaC-mNeo and YaaT-mNeo. (**A**) Heat maps of single-molecule localization of replication proteins in a medium-size *Bacillus subtilis* cell. The distribution of tracks is indicated by a colour shift from yellow (low probability) to black (highest probability). (**B**) Jump distance analyses of ExoR-mNeo and RNase HII-mNeo. The two Rayleigh fits display a two-population fit (static in blue, mobile in red and added fits in black). (**C**) Bubble blots show diffusion constants of replication proteins and fraction sizes for static and mobile molecules.

**Figure 3 cells-13-01381-f003:**
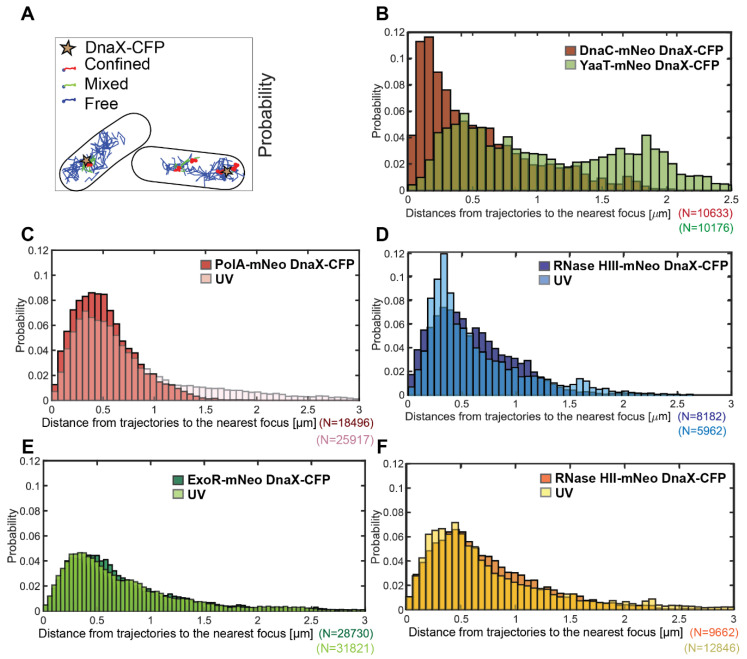
Distance measurement of potential replication proteins to DnaX-CFP (replication fork). Panel (**A**) shows an example of the confinement analyses of RNase HII-mNeo. The replication forks are marked with a star. Confined tracks are red, transition tracks are green and free tracks are blue. The marker for the replication fork is DnaX-CFP, expressed from the original gene locus. At least 50 cells were measured. The number of tracks measured is indicated (N). The probability of detection close to DnaX-CFP foci is shown in relation to the distance in micrometers (µm). In Panel (**B**), the replication protein DnaC (brown) and the negative control strain YaaT are displayed (green). (**C**–**F**) show tested proteins under normal conditions (darker colours) in overlay with UV-induced DNA damage (brighter colours).

**Figure 4 cells-13-01381-f004:**
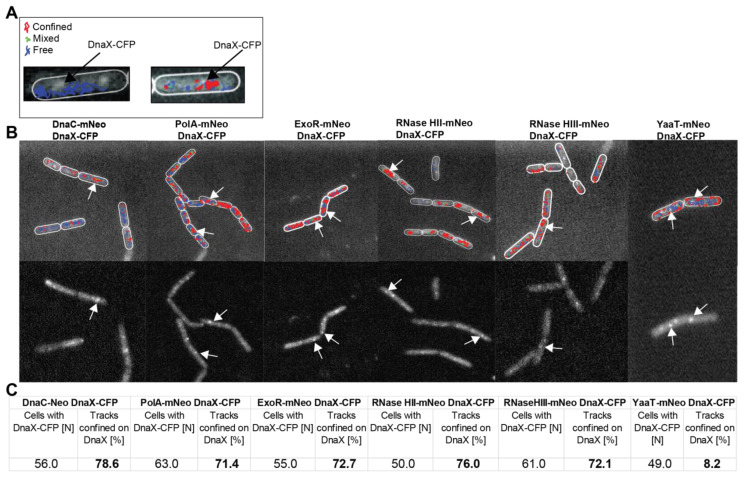
Confinement analyses of replication-associated proteins relative to DnaX-CFP foci. Panel (**A**) shows exemplarily a cell (DnaX-CFP signal visible with white arrows) with free tracks and a cell with free (blue), confined (red) and mixed (green) tracks. Panel (**B**) (upper part) shows an overlay of confinement analyses of DnaC-mNeo, PolA-mNeo, ExoR-mNeo, RNase HII-mNeo, RNase HIII-mNeo and YaaT-mNeo in relation to the foci of the replication fork (white arrows show exemplary DnaX-CFP, lower part). (**C**) More than 50 cells per strain were analysed. The cells with visible DnaX-CFP signals are indicated (N). Cells with confined tracks at the replication fork are indicated in percent [%]. The tracks are divided into confined (red), in transition (green) and free (blue).

**Figure 5 cells-13-01381-f005:**
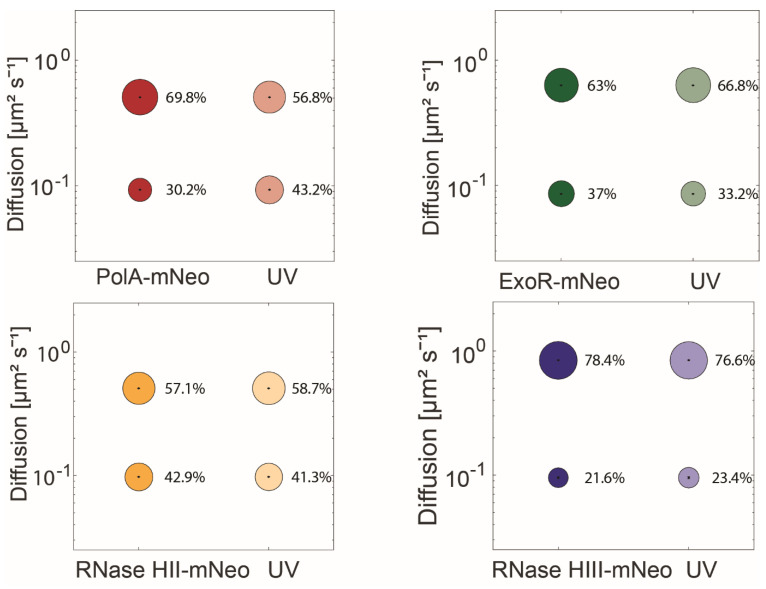
Analyses of protein dynamics via single-molecule tracking with or without UV treatment. Bubble blots show diffusion constants of replication proteins and fraction sizes for static and mobile molecules with and without treatment with UV-light. The dynamics are divided into a static and mobile population.

**Figure 6 cells-13-01381-f006:**
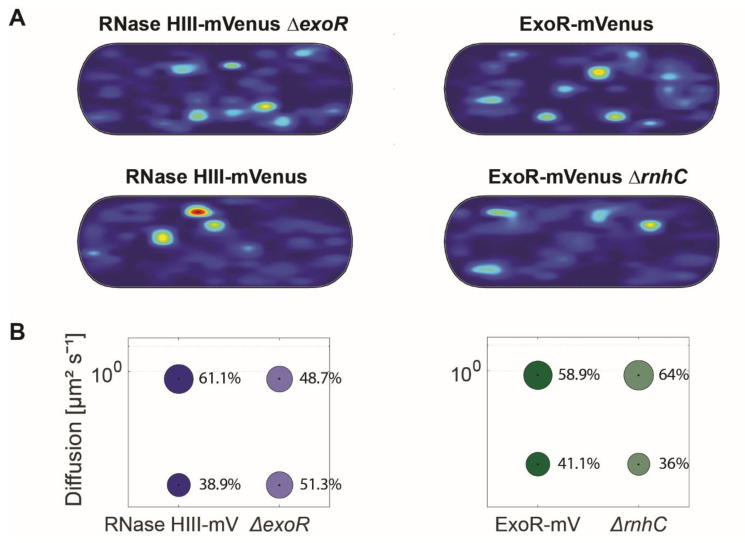
Analyses of protein dynamics via single molecule tracking of RNase HIII and ExoR in deletion backgrounds. (**A**) Confinement maps of RNase HIII-mV and ExoR-mV in comparison to corresponding deletion background. Plots of heat maps of confined tracks projected into a standardized *B. subtilis* cell. (**B**) Bubble blots show diffusion constants of replication proteins and fraction sizes for static and mobile molecules and in the absence of corresponding proteins.

**Figure 7 cells-13-01381-f007:**
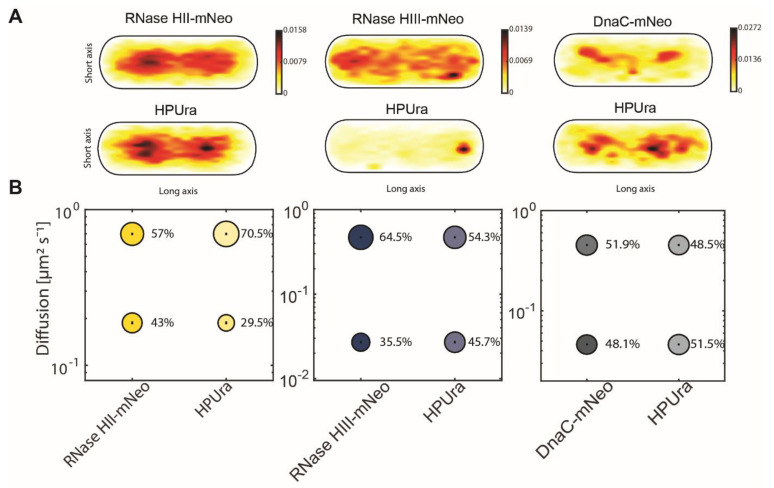
Analyses of protein dynamics under treatment of HPUra via single-molecule tracking. (**A**) Heat maps of single-molecule localization of RNase HII RNase HIII and DnaC with and without HPUra in a medium-size Bacillus subtilis cell. The distribution of tracks is indicated by a colour shift from yellow (low probability) to black (highest probability). (**B**) Bubble blots show diffusion constants of RNase HII-mNeo, RNase HIII-mNeo and DnaC-mNeo and fraction sizes for static and mobile molecules under the treatment of HPUra.

**Figure 8 cells-13-01381-f008:**
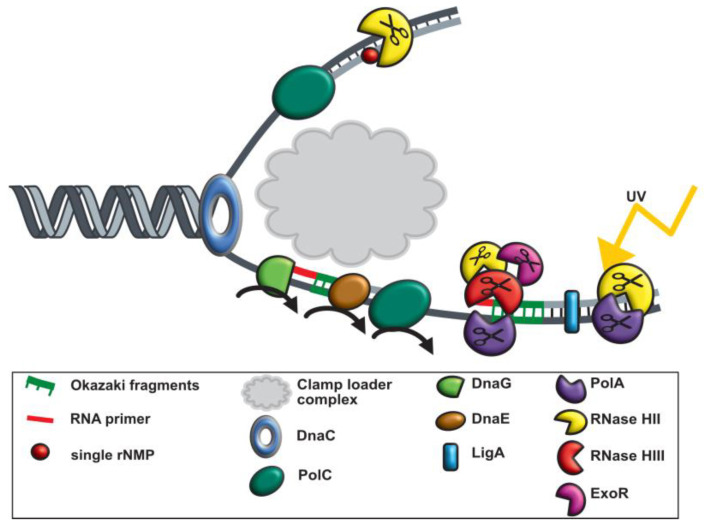
Graphical illustration of replication in *Bacillus subtilis* during UV stress. At the replication fork, a DNA helicase (DnaC) precedes the DNA synthesis machinery and unwinds the exist duplex parental DNA in cooperation with the single stranded binding proteins (SSB). On the leading strand (5 to 3 direction), replication proceeds continuously via the replisome. In contrast, on the lagging strand, DNA replication is performed discontinuously by synthesizing and assembling short Okazaki fragments. DNA primase (DnaG) is required for the formation of RNA primers. During DNA template replication, an RNA primer is removed either by the 5-bis-3 exonuclease activity of ExoR and Okazaki fragments are matured via PolA, RNase HII and RNase HIII. Afterwards DNA ligase LigA fuses matured Okazaki fragments.

**Table 1 cells-13-01381-t001:** List of square displays meant (SQD) analyses.

Figure 2								
	PolA-mNeo	ExoR-mNeo	RNase HII-mNeo	RNase HIII-mNeo	DnaC-mNeo	YaaT-mNeo		
D1 (static) µm^2^ s^−1^	0.09 ± 0.0	0.09 ± 0.0	0.10 ± 0.0	0.13 ± 0.001	0.05 ± 0.001	0.05 ± 0.001		
D3 (mobile) µm^2^ s^−1^	0.51 ± 0.001	0.63 ± 0.001	0.50 ± 0.001	0.82 ± 0.001	0.4 ± 0.002	0.62 ± 0.002		
Static %	29.9 ± 0.001	35.2 ± 0.001	43.0 ± 0.001	23.2 ± 0.001	37.6 ± 0.001	53.0 ± 0.00		
Mobile %	70.1 ± 0.001	64.8 ± 0.001	57.0 ± 0.001	76.8 ± 0.001	62.4 ± 0.001	47.0 ± 0.00		
R2	1	0.999	1	1	0.999	0.999		

Figure 5								
	PolA-mNeo	UV	ExoR-mNeo	UV	RNase HII-mNeo	UV	RNase HIII-mNeo	UV
D1 (static) µm^2^ s^−1^	0.09 ± 0.0	0.09 ± 0.0	0.09 ± 0.0	0.09 ± 0.0	0.10 ± 0.0	0.10 ± 0.0	0.10 ± 0.003	0.10 ± 0.003
D3 (mobile) µm^2^ s^−1^	0.51 ± 0.001	0.51 ± 0.001	0.63 ± 0.001	0.63 ± 0.001	0.51 ± 0.002	0.51 ± 0.002	0.84 ± 0.004	0.84 ± 0.004
Static %	30.2 ± 0.001	43.2 ± 0.001	37 ± 0.001	33.2 ± 0.001	42.9 ± 0.002	41.3 ± 0.002	21.6 ± 0.003	23.4 ± 0.003
Mobile %	69.8 ± 0.001	56.8 ± 0.001	63 ± 0.001	66.8 ± 0.001	57.1 ± 0.002	58.7 ± 0.002	78.4 ± 0.003	76.6 ± 0.003
R2	1	0.999	1	1	1	1	0.999	0.996

Figure 6								
	ExoR-mV	ExoR-mV Δ*rnhC*	RNaseHIII-mV	RNaseHIII-mV Δ*exoR*				
D1 (static) µm^2^ s^−1^	0.09 ± 0.001	0.09 ± 0.001	0.05 ± 0.001	0.05 ± 0.001				
D3 (mobile) µm^2^ s^−1^	0.85 ± 0.005	0.85 ± 0.005	0.84 ± 0.001	0.84 ± 0.001				
Static %	41.1 ± 0.267	36 ± 0.178	38.9 ± 0.181	48.7 ± 0.264				
Mobile %	58.9 ± 0.267	64 ± 0.178	61.1 ± 0.181	51.3 ± 0.264				
R2	0.995	0.999	0.994	0.992				

Figure 7								
	RNaseHII-mNeo	HPUra	RNaseHIII-mNeo	HPUra	DnaC-mNeo	HPUra		
D1 (static) µm^2^ s^−1^	0.19 ± 0.002	0.19 ± 0.002	0.03 ± 0.001	0.03 ± 0.0	0.05 ± 0.0	0.05 ± 0.0		
D3 (mobile) µm^2^ s^−1^	0.7 ± 0.005	0.7 ± 0.005	0.47 ± 0.004	0.47 ± 0.004	0.45 ± 0.003	0.45 ± 0.003		
Static %	43 ± 0.006	29.5 ± 0.006	35.5 ± 0.003	45.7 ± 0.003	48.1 ± 0.002	51.5 ± 0.002		
Mobile %	57 ± 0.006	70.5 ± 0.006	64.5 ± 0.004	54.3 ± 0.004	51.9 ± 0.002	48.5 ± 0.002		
R2	0.999	0.999	0.972	0.972	0.999	0.999		

## Data Availability

All relevant data are shown within this study. Upon reasonable request, raw data for single-molecule tracking movies can be obtained from the corresponding authors.

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
