# Peer review of "Visual Evidence for the Recruitment of Four Enzymes with RNase Activity to the Bacillus subtilis Replication Forks"

_cells, 2024, doi:10.3390/cells13161381_

Round 1

Reviewer 1 Report

Comments and Suggestions for Authors

The authors studied the recruitment of PolA and three RNases (ExoR, RNase HII, and RNase HIII) into Bacillus subtilis replisome using in vivo single molecule tracking. The scientific objectives of this study were clearly defined, and the experiments were carefully planned and performed with great care. Indeed, a state-of-the-art methodology provides a convincing argument that all four enzymes are recruited at replication forks; the described experiments allowed authors to capture events of transitions from free diffusion to DNA binding and release from DNA binding, as pointed out in the Results section (lines 198-199).

Abstract (lane 20). Please change HPUra for 6(p-hydroxyphenylazo)uracil.

Introduction. (lane 36) deoxyribonucleotides instead desoxy ribonucleotides.

(lane 37) give RTP in full (Ribonucleoside TriPhosphate)

(lane 38) Please correct this; ribonucleoside triphosphates can be incorporated into the growing primer strand. Ribonucleoside monophosphates are present in the newly synthesized strand due to misincorporation events.

(lane 44) use 3’-5’, 5’-3’ or 3’->5’, 5’->3’ (see lane 48) instead 3’ to 5’ or 5’ to 3’

Results (Fig. S1) Please provide the source of mNeonGreen antibodies and the source of the protein markers.

(lanes 94-95) Have the Authors analyzed the kinetics of loss of a chromosome-integrated plasmid? Are they sure that the construct produced the fused enzyme during the experiment?

(lane 98) the term “expression” is reserved for genes

(lane 226) “peak” instead of “peat”

In cells with arrested replication (UV treatment), a substantial shift of RNase HIII and PolA to replication forks was observed (Fig. 3). This is contradicted by the conclusion presented in the following Results subsection: “PolA, but not RNase HIII or HII, shows a change in mobility in response to UV light-induced DNA damage” (line 261-262). Please explain this discrepancy.

The experiment with DNA replication arrested by HPUra (a PolC inhibitor) is elegant and shows differences in the engagement of RNase HII and HIII (Fig. 7).

(line 334) use straightforward instead straight forward

Discussion. The results obtained were thoroughly reviewed. This chapter is well-organized and easy to follow. The final paragraph presenting a step-by-step scenario of B. subtilis DNA replication is compelling and clarifies the function of analyzed enzymes. I strongly agree with the Authors that studies on B. subtilis DNA replication extend our textbook knowledge concerning this vital process.

(lane 355) Pol I instead of Pol 1.

(lane 385) DNA methyltransferases instead of DNA methylases.

(lanes 404-405)The steric gate responsible for preferential incorporation of deoxyribonucleotides over ribonucleotides should be discussed in more detail, as model enzymes such as Eco PolA or T7 PolA have been extensively studied in this respect.

Fig. 8. The graphical illustration should be of better quality. The symbols depicting particular enzymes should be more distinctive and in brighter colours.

Materials and methods. (lane 480) add a reference to pSG1164.

References. Use italics for genes and bacterial strains. Use a uniform style for journal titles.

Line 552, add the journal title to the ref. 1, Wiley Interdiscip. Rev. RNA

Line 553, add the journal title to the ref. 2, Int. J. Microbiol.

Line 555, add the journal title to the ref. 3, Cold Spring Harb. Perspect. Biol.

Line 557 add the journal title to the ref. 4, FEMS Microbiol. Rev.

Line 559, add the journal title to the reference. 5, Cell

Line 560, add the journal title to the ref. 6, Nat. Struct. Mol. Biol.

Line 570, add the journal title to the ref. 11, Mol. Gen. Genet.

Line 578, add the journal title to the ref. 15, Biochemistry

Line 595, add the journal title to the ref. 23, Proc. Natl. Acad. Sci. USA

Line 600-601. Add the journal title to the ref. 26, Nucl. Acids Res.

Line 624, add the journal title to the ref. 36, Front. Microbiol.

Line 643, BMC Biology instead of BMC Biology (ref. 44)

Line 659, add the journal title to the ref. 51, J. Bacteriol.

Line 663, add the journal title to the ref. 32, PloS One

Line 669, BMC Biology instead of BMC Biology (ref. 55)

Conclusion (for editor): I found the present submission interesting, well-written, and worthy of publication in your respected journal.

Reviewer 2 Report

Comments and Suggestions for Authors

In this manuscript by Hinrichs and Graumann the localization of RNase activity possessing PolA, ExoR, RNase HII, and RNase HIII, at the replication fork in Bacillus subtilis were analyzed using single molecule tracking. The authors employed a systematic approach by first analyzing the localization of these enzymes followed by measuring distance of tracks to the replication forks. The authors went further to analyze how these enzymes respond to UV induced DNA damage and inhibition of replication by HPUra. Overall this is a well-written manuscript and an exciting set of experiments.

Reviewer 3 Report

Comments and Suggestions for Authors

Dear authors your manuscript titled “Visual evidence for the recruitment of four enzymes with RNase activity to the Bacillus subtilis replication forks” investigates a particular aspect of the bacterial mechanisms involved in one stage of DNA replication.

The introduction is essential but accurately describes the behaviour in Bacillus subtilis, the subject of the research conducted.

The materials and methods are well described and all steps from the construction of the strain to the culture of the bacteria and the other techniques used. The pictures, tables and figures are of high quality with well-explanatory captions.

The results are well described and the considerations comment fully on the results obtained.

There are no comments to be made.
